# Factors influencing the use of emergency contraceptives among reproductive age women in the Kwadaso Municipality, Ghana

Daniel Sarpong Yeboah[1], Maxwell Afranie Appiah[2]*, Grace Billi Kampitib[3]

1 Obstetrics and Gynaecology Department, Kumasi South Hospital, Ghana Health Service, Kumasi, Ashanti Region, Ghana, 2 District Health Directorate, Ghana Health Service, Obuasi East, Obuasi, Ashanti Region, Ghana, 3 Municipal Health Directorate, Ghana Health Service, Kwadaso Municipal, Kwadaso, Ashanti Region, Ghana

* 2014mappiah@uhas.edu.gh

## Abstract

### Background

Unintended pregnancy leads to unsafe abortion, which is one of the commonest causes of maternal deaths in developing countries including Ghana. Lots of unintended pregnancies can be avoided using emergency contraceptives (EC). Emergency contraceptives are mostly used after unprotected sexual intercourse and have a ninety-nine percent chance of preventing unintended pregnancy when taken correctly. However, unlike other modern contraceptives such as condoms, emergency contraceptives cannot prevent sexually transmitted infections.

### Objectives

This study aimed at assessing the factors influencing the use of emergency contraceptives among reproductive-age women in the Kwadaso Municipality, Ghana.

### Methods

A community-based descriptive cross-sectional study design was conducted in three sub-municipalities of the Kwadaso Municipality. A multistage sampling method was used to select 312 women in their reproductive age within households. A simple random sampling method was first used to select the sub-municipalities (Kwadaso Central, Asuoyeboah, and Agric-Nzema). Participants were selected from households through a systematic sampling procedure and responses were solicited from women who consented to participate in the study. The selection was strictly dependent on the number of eligible women in a household, that is, in an event where more than one woman was found in a household, a simple random sampling method was used to select only one woman from that household. STATA 15.0 was used to analyse the data. Binary logistic regression was used to find the adjusted estimates and associations between EC use and the exposure variables. P-values $\leq 0.05$ were considered statistically significant at 95% Confidence Interval (CI).

**Data Availability Statement:** All relevant data are within the paper and its Supporting Information files

**Funding:** The authors received no specific funding for this work.

**Competing interests:** The authors have declared that no competing interests exist.

## Results

The findings showed that 79.67% of the women had ever used EC. Amongst them, 59.83% used EC following unexpected unprotected sex, and 24.69% used EC following failed coitus interruptus. Women's attitude towards EC (AOR = 8.52, *p<0.001*), religion (AOR = 4.56, *p = 0.004*), and monthly income (AOR = 0.29, *p = 0.030*) were found to have significant influence on their use of EC.

## Conclusion

The level of EC use among the women was high. Women's attitude towards EC, religion, and monthly income were the major factors influencing the use of EC. Thus, strategies to promote EC use should emphasize on addressing the attitude of women towards EC through sex education in schools, various religious institutions, and the community at large with the services of health authorities and support from governmental and non-governmental organizations whose focus is to address the need for reproductive health services in order to reduce the misconception regarding the use of EC.

## Introduction

Emergency contraceptives (ECs) can be used to prevent unintended pregnancies and unsafe abortions. Distinct from other regular contraceptives, ECs are mostly used after unprotected sexual intercourse [1]. ECs are effective when used shortly within seventy-two (72) hours after unprotected sex [2]. Generally, ECs can be categorized into two namely, emergency contraceptive pills (ECPs) and intrauterine contraceptive devices (IUDs) [3]. The ECPs are further grouped into two; combined oral contraceptives pills (COCs), which contain oestrogen and progestin, and progesterone-only pills (POPs) containing only progesterone [2, 3]. ECPs can prevent unintended pregnancies when used within 72 and 120 hours of unprotected sexual intercourse [4]. The IUDs on the other hand can be effective when inserted within five (5) days after sexual intercourse [2].

Emergency contraceptives can decrease the risk of unintended pregnancy by 75% to 99% when taken within 72 hours of sexual intercourse [1, 5]. ECPs have 75% to 85% chances of preventing unintended pregnancies whilst the combination of both ECPs and IUDs can prevent about 99% of unintended pregnancies [6]. ECs are cost-effective [7] with prices ranging from US$ 1.20 to US$ 5.78 [8]. Also, ECs are medically safe, yet accompanied by minor side effects such as nausea, vomiting, menstrual irregularity, fatigue, slight irregular vaginal bleeding, and breast tenderness [4, 7].

Global estimation shows that 222 million women who want to prevent pregnancy are not accessing effective, modern methods of contraception. As a result, approximately 86 million unintended pregnancies, 20 million unsafe abortions, and 33 million unexpected deliveries occur in the world yearly [9]. Additionally, an estimated 21.6 million unintended pregnancies occur in Africa, of which close to 38% of them end in induced (safe and unsafe) abortions every year [10]. Also, the highest proportion of all unsafe abortions occur in developing countries, especially in Africa [11]. Unsafe abortion can lead to further complications such as haemorrhage, infections, infertility, or even death [12]. Studies have shown that female adolescents and unmarried women are often highly vulnerable to sexual coercion and violence leading to unintended pregnancies and abortions [3, 13]. More so, adolescent and young adult women

fall under the sexually active age group and form a high-risk group for unintended pregnancies, since the likelihood for them to engage in sporadic pre-marital sex is high [13]. Research suggests that early involvement in sexual activity and low contraceptive usage are some known reasons that contribute to unintended pregnancies in Africa [14, 15].

In Ghana, one key strategy for controlling the population is through family planning methods. The use of EC has become an integral part of family planning services to prevent conception following unprotected sex or a contraceptive accident like condom breakages, slippage of the diaphragm, and failed coitus interruptus [16]. Presently, knowledge regarding the use of ECs is almost universal in Ghana, however, the current estimate indicates that 30% of married women in the country have an unmet need for family planning services, with 17% of them having an unmet need for spacing, and 13% have an unmet need for limiting [17]. Also, the rate at which contraceptives are utilized among all women is very low in the country with a national usage rate of 27% [18]. Similarly, the use of contraceptives in the Ashanti region, where Kwadaso Municipality is located, is equally low with a usage rate of 27% [18].

Since the inception of EC in Ghana, some studies have been conducted to identify factors influencing the use of EC, however, the focus of most of these studies are on female college or university students [19, 20], with limited studies focusing on communities in the country and among women of reproductive age. This study thus investigated the factors influencing the use of EC among women of reproductive age in the Kwadaso Municipality of Ghana. In addition to complementing existing literature on reproductive health, having an understanding of the factors influencing EC use at community level is crucial to implementing population control measures of reducing unintended pregnancies and achieving a population growth reduction rate of 1.5% [21]. This will culminate in the improvement of the country's labour force by ensuring maximum utilization of resources. Thus, as part of the country's policy and decision-making process, the outcome of this study would serve as a standard to address reproductive health goals related to the use of EC in the Kwadaso Municipality and the nation at large.

## Materials and methods

### Study design and setting

A community-based descriptive cross-sectional study design was used to conduct the study in the Kwadaso Municipality. The Kwadaso Municipality is situated in the western part of Kumasi in the Ashanti Region, Ghana. The Municipality shares boundary with the Atwima Nwabiagya District to the North, Bantama Sub-Metro to the South, Nyiaeso Sub-Metro to the East and Atwima Kwanwoma District to the West. It has an estimated population of 251, 215 with a growth rate of 2.3%. The municipality has five (5) health demarcated sub-municipalities namely, Agric-Nzema, Apatrapa–Nyankyerenease, Asuoyeboah, Kwadaso Central, and Tanoso-Takyiman. There are 16 communities in the entire municipality which include; Apatrapa-Poku-Krom, Nyankyerenease, Topre, Asuoyeboah, Denchenmuoso, Apire, Atwima-Takyiman, Nwamase, Kwadaso, Nzema, Edwenase, Tanoso, Ohwimase, Kokode, and Atwima Amanfrom.

### Sampling procedure

The study included 312 women of reproductive age (15 to 49 years) in the Kwadaso Municipality. The desired sample size was obtained using Cochran's formula; $n = z^2pq/d^2$. The rate of EC use was estimated at 27% [18] with a Z-score of 1.96 at a 95% confidence level, a marginal error of 5% and a non-response rate of 5%. A multistage sampling method was used to select participants for the study. The first stage involved the use of a simple random sampling technique through a toss to select three (3) sub-municipalities (i.e., Kwadaso Central, Asuoyeboah,

and Agric-Nzema) out of the five (5) sub-municipalities that form the Kwadaso Municipality. A systematic sampling procedure was used at the second stage to selected houses within the three sub-municipalities from which to select a participant. The houses were numbered and the first house was selected using a random number generator. If a selected house comprised of multiple households (i.e. two or more), one household was randomly selected and only one woman in the reproductive age within that household was selected. In an instance where there was more than one woman in the same household within the reproductive age, selection was done using simple random sampling technique, that is, all eligible members were assigned a number from which one number was selected at random by a neutral person. However, participation was based on the individual consent. This was done until the sample size of 312 was reached. Based on the estimated population for 2019 for each of the areas included in the study, 167 women were sampled from Kwadaso Central, 91 women were sampled from Agric-Nzema, and 54 women were sampled from Asuoyeboah sub-municipalities. Proportionate sampling ensured that each sub-municipality was fairly represented in the sample population.

## Data collection tool

Data on participants were collected using a structured questionnaire. The questionnaire was categorized into four sections including participants' sociodemographic and economic characteristics, knowledge, attitude, and practices of emergency contraception (S1 File). The questionnaire was adapted from existing literature [22], and pretested on twenty-five (25) market women in Bantama, a suburb of Kumasi Metropolitan. This was carried out to validate the questionnaire and also establish whether or not the questions could be easily understood and exclude questions that had no bearings with the study. The questionnaire items for knowledge and attitude were tested for reliability. The results showed a Cronbach alpha (α) value of 0.73, suggesting an acceptable internal consistency between the measurement constructs and their questionnaire items.

## Data collection procedure

The questionnaires were administered to the participants through face-to-face interview method, from November 2019 to January 2020, by the principal investigators and two trained public health officers. The questions were written in English since it is the official language of Ghana. However, for the uneducated, questions were interpreted in the participant's local dialect (Asanti Twi) and where necessary, samples and images of ECs were made available to enable them to respond correctly to the questions. Each interview took an average of twenty minutes.

## Data analysis

Data were entered into Microsoft Excel 2016 version and imported to STATA 15.0 (StataCorp, College Station, Texas 77845 USA) for analysis. A descriptive analysis was conducted on respondents' socio-demographic and economic characteristics, knowledge, attitude and practices of emergency contraception. Respondent's level of knowledge was assessed using ten questions that consisted of general knowledge about EC and its use including the following: "Can EC prevent pregnancy", "Is EC a regular method of contraception", "What is the recommended time for taking EC", "Are there side effects of EC", "Where can EC be accessed", "Can EC prevent STI" and others. Respondents were given "yes", "no", or "I don't know" response options to these questions. All correct responses were scored one point whilst incorrect / I don't know responses were scored zero point. An overall knowledge score was calculated by adding up the scores for each respondent across all ten questions. The cumulative score of the

ten questions ranged from 4 to 10 points for a given respondent, and then this distribution was dichotomized at a mean score value of 7.0 (SD = 1.5). Respondents whose score were equal or greater than the mean were considered as having "good knowledge" while those with scores less than the mean score were considered as having "poor knowledge" about EC and its use [1].

Similarly, attitude towards EC was assessed using ten questions. Responses to questions related to attitude were graded on a 3-point Likert scale; an agreement scale of agreed, disagreed or not sure. By scoring three points for correct answer and one point for incorrect answer following the Likert's scale, an overall attitude score was determined for each respondent by adding up the scores across the ten attitude questions. The maximum score possible was 30 points, with a mean score value of 22.8 (SD = 3.3). Respondents who had a score greater than or equal to the mean score were considered as having "positive attitude" while those with scores less than the mean score were considered as having "negative attitude" towards EC [1].

Binary logistic regression analysis was conducted to identify factors influencing the utilization of emergency contraception among the respondents. The outcome variable "Ever used EC" was transformed into a dummy variable (yes = 1, no = 0). Thus, the logistic regression analysis was applied to "yes" responses of the outcome variable to assess the factors that influenced participants' use of EC. The significant variables in the univariate analysis were further examined using stepwise binary logistic regression in order to identify the significant predictors after controlling other variables. P-values $\leq$ 0.05 were considered statistically significant at a 95% Confidence Interval (CI).

## Ethics approval

Ethical clearance and study approval were obtained from the Committee on Human Research, Publications, and Ethics of the Kwame Nkrumah University of Science and Technology (KNUST) (Ref No:CHRPE/AP/617/19). Written informed consent was obtained from all participants, however, assent was obtained from parents or guardians of participants with ages below 18 years (S2 File). Participants' confidentiality was maintained at all times. To further maintain anonymity, no identifying information was collected and responses were tagged using study code numbers. Participation was voluntary and the participants were informed that they could withdraw from the study at any stage of the interview, if they so desired, without any penalty or explanation.

## Results

### Sociodemographic and economic characteristics

The socio-demographic and economic characteristics of participants are shown in Table 1. Considering the sociodemographic status, the age range of the respondents was from 18 to 35 years with a median age of 24 years, a mean age of 25.56 years and a standard deviation of 4.46 years. Majority of the respondents (48.08%) were within the age group 25 years and greater followed by age group 20 to 24 years (45.83%). Most of them (85.58%) were of Christian background. Also, the majority of the respondents (65.71%) were single. Thirty-eight percent (38.14%) of them had completed tertiary education with a Bachelor's degree, 28.85% had completed Senior high school and 8.33% of them had completed Junior high school.

Regarding the economic status of the respondents, 54.49% of them worked in the last month prior to the study period. Among those who did not work, the majority of them (47.13%) stated that there was no work available for them, others (36.62%) were students, and some of them (2.11%) were sick. Furthermore, 90.00% of those who worked in the last month prior to the study received regular wages or salary as a form of payment. Also, 37.06% and

**Table 1. Sociodemographic and economic characteristics of respondents.**

| Variables | Frequency | Percentage |
|---|---|---|
| *Sociodemographic* | | |
| **Age** | | |
| ≤ 19 years | 19 | 6.09 |
| 20 to 24 years | 143 | 45.83 |
| ≥ 25 years | 150 | 48.08 |
| **Religion** | | |
| Christianity | 267 | 85.58 |
| Muslim | 45 | 14.42 |
| **Highest level of education** | | |
| No formal education | 0 | 0.00 |
| Primary | 0 | 0.00 |
| Junior High school | 26 | 8.33 |
| Completed Senior High school | 90 | 28.85 |
| Vocational degree or certificate | 77 | 24.68 |
| Bachelor's degree | 119 | 38.14 |
| Graduate or advanced professional | 0 | 0.00 |
| **Marital Status** | | |
| Single | 205 | 65.71 |
| Married | 107 | 34.29 |
| *Economic characteristics* | | |
| **Working for pay in the last month** | | |
| Yes | 170 | 54.49 |
| No | 142 | 45.51 |
| **If no, main reason for not working (n = 142)** | | |
| No work available | 67 | 47.18 |
| Seasonal inactivity | 8 | 5.63 |
| Student | 52 | 36.62 |
| Household / family duties | 12 | 8.45 |
| Infirmity / sickness | 3 | 2.11 |
| **How were you paid for your work? (if unemployed use last job) (n = 170)** | | |
| Regular wages or salary | 153 | 90.00 |
| Casual labour (hourly / daily) | 17 | 10.00 |
| **Monthly income (n = 170)** | | |
| Less than or equal GH₵ 500 | 46 | 27.06 |
| GH₵ 501 to 1000 | 49 | 28.82 |
| GH₵ 1001 to 1500 | 63 | 37.06 |
| GH₵ 1501 to 2000 | 8 | 4.71 |
| More than GH₵ 2000 | 4 | 2.35 |

28.82% of them had monthly income ranging from GH₵ 1001 to GH₵ 1500 and GH₵ 501 to GH₵ 1000 respectively.

## Knowledge about emergency contraception

The majority of the respondents (96.15%) had heard of EC (Table 2). Amongst them, proges-terone-only pills (e.g., Lydia, Postinor2) were the most known EC (92.00%), followed by com-bined oral contraceptive pills (58.67%) and intrauterine copper devices (54.67%). Also, a few of them (2.33%) were aware of EC, yet did not know what it actually was (Fig 1). A greater

**Table 2. Respondents' knowledge about emergency contraception.**

| Variables | Frequency | Percentage |
|---|---|---|
| **Have you ever heard of emergency contraception?** | | |
| Yes | 300 | 96.15 |
| No | 12 | 3.85 |
| **If yes, is EC recommended as a regular contraceptive method? (n = 300)** | | |
| Yes | 48 | 16.00 |
| No | 184 | 61.33 |
| I don't know | 68 | 22.67 |
| **Can ECs be used to prevent unwanted pregnancy? (n = 300)** | | |
| Yes | 284 | 94.67 |
| No | 16 | 5.33 |
| **Where can EC be obtained (n = 300)** | | |
| Pharmacy / health facility | 296 | 98.67 |
| Supermarket / any shop | 4 | 1.33 |
| I don't know | 0 | 0.00 |
| **Are there side effects associated with the use of EC? (n = 300)** | | |
| Yes | 279 | 93.00 |
| No | 21 | 7.00 |
| **What are some of the side effects of EC (n = 249)** | | |
| Menstrual irregularity | 128 | 51.41 |
| Breast tenderness | 25 | 10.04 |
| Ectopic pregnancy | 24 | 9.64 |
| Vomiting | 10 | 4.02 |
| Drowsiness | 16 | 6.43 |
| Delay fertility | 22 | 8.84 |
| Spotting | 8 | 3.21 |
| Nausea | 16 | 6.43 |
| **Recommended duration for taking EC** | | |
| Within 12 hours | 17 | 5.67 |
| Within 24 hours | 40 | 12.67 |
| Within 48 hours | 22 | 7.33 |
| Within 72 hours | 209 | 68.00 |
| I don't know | 24 | 6.33 |
| **Can emergency contraception prevents STI (n = 300)** | | |
| Yes | 6 | 2.00 |
| No | 281 | 93.67 |
| I don't know | 13 | 4.33 |
| **Knowledge about EC (Summary Index)** | | |
| Poor knowledge | 114 | 38.00 |
| Good knowledge | 186 | 62.00 |

number of the respondents (81.00%) who had heard of EC had their source of information from a health professional (Fig 2) (S1 Fig). Also, the majority (61.33%) knew that EC was not recommended as a regular contraceptive method. Conversely, 22.67% of them did not know whether EC was recommended as a regular method of contraception or not. Most of them (94.67%) knew that EC can be used to prevent unwanted pregnancies. Similarly, the majority of them (68.00%) knew that the recommended duration for EC use was within 72 hours after unprotected sex. Also, 6.33% of them did not know the time limit for taking emergency

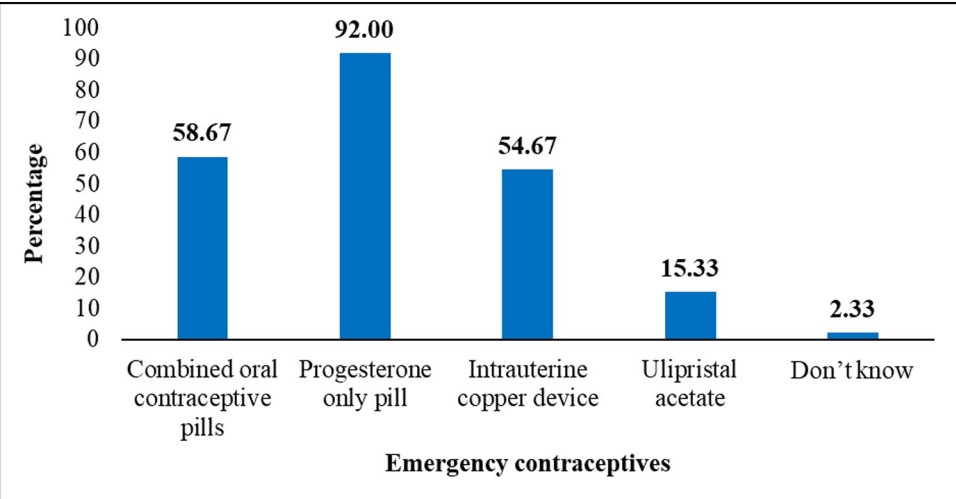

**Fig 1. Respondents' knowledge on the methods of emergency contraception (n = 300).**

contraceptives. Furthermore, most of the respondents (98.67%) knew that ECs could be obtained from the pharmacy and at the health facility. Amongst those who had heard about EC, the majority of them (93.00%) were aware that ECs had an associated side effect. However, 89.25% of them could identify some side effects of EC. Menstrual irregularities, breast tenderness, ectopic pregnancy, vomiting, drowsiness, delay fertility, spotting, and nausea were identified as the side effects associated with the use of EC. Additionally, most of them (93.67%) knew EC cannot prevent sexually transmitted infection (STI). With regards to the overall level of knowledge, 62.00% of respondents had a good knowledge about EC (Table 2).

## Attitude towards emergency contraception

Table 3 presents the respondents' attitude towards the use of emergency contraceptives. Forty-eight percent (48.67%) of the respondents agreed to the statement "The provision of EC to women would encourage promiscuity and hence increase the prevalence of HIV / AIDS and

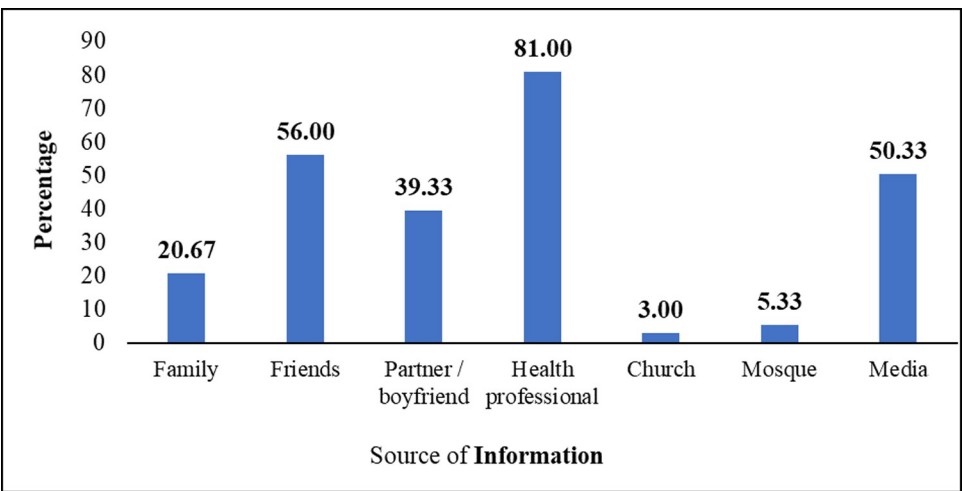

**Fig 2. Respondents' source of information about emergency contraception (n = 300).**

**Table 3. Respondents' attitude towards the use of emergency contraceptives (n = 300).**

| Variables | Agree | | Not sure | | Disagree | |
|---|---|---|---|---|---|---|
| | No. | % | No. | % | No. | % |
| The provision of EC to women would encourage promiscuity hence increase the prevalence of HIV / AIDS and other STIs | 146 | 48.67 | 31 | 10.33 | 123 | 41.00 |
| The provision of EC would discourage compliance with other contraceptive methods | 156 | 52.00 | 63 | 21.00 | 81 | 27.00 |
| Repeated use of EC poses a health risk | 251 | 83.67 | 49 | 16.33 | 0 | 0.00 |
| EC should be prescribed for a client to have on hand prior to an episode of unprotected sexual intercourse | 190 | 63.33 | 56 | 18.67 | 54 | 18.00 |
| EC should be available without prescription | 162 | 54.00 | 59 | 19.67 | 79 | 26.33 |
| EC should be easily made accessible to all females | 239 | 79.67 | 33 | 11.00 | 28 | 9.33 |
| EC should be used regularly to prevent unwanted pregnancy | 108 | 36.00 | 32 | 10.67 | 160 | 53.33 |
| EC is a safe method of preventing unplanned pregnancy | 173 | 57.67 | 68 | 22.67 | 59 | 19.67 |
| Will use EC in the future when the need arises | 203 | 67.67 | 40 | 13.33 | 57 | 19.00 |
| Will advise family members and friends to use EC | 177 | 59.00 | 62 | 20.67 | 61 | 20.33 |
| **Attitude towards EC (Summary Index)** | | | No. | | | % |
| Negative attitude | | | 141 | | | 47.00 |
| Positive attitude | | | 159 | | | 53.00 |

No. = number; % = percentage

other STIs". Also, the majority of them (52.00%) agreed that "the provision of EC would discourage compliance to other contraceptive methods". Similarly, the majority of them (83.67%) agreed that "repeated use of EC pose a health risk" and none of them was in disagreement with the statement. Additionally, 63.34% of the respondents agreed that "EC should be prescribed for a client to have on hand prior to an episode of unprotected sexual intercourse". Likewise, 54.00% of them agreed that "EC should be available without a prescription" and 26.33% of them disagreed with such statement. More so, the majority of them (79.67%) agreed that "EC should be easily made accessible to all females". Also, most of them (53.33%) disagreed that "EC should be used regularly to prevent unintended pregnancy". Furthermore, the majority of respondents (67.67%) agreed that they will use EC in the future when the need arises, and 59.00% of them agreed that they will advise family members and friends to use EC. Concerning the overall level of attitude toward EC, more than half of respondents (53.00%) showed a positive attitude towards EC (Table 3).

## Use of emergency contraception

Most of the respondents (79.67%) indicated that they had ever used emergency contraceptives. Amongst them, 59.83% noted that they used EC because they had unprotected sex. Others (24.69%) also noted failed coitus interruptus as a reason for using emergency contraceptives. Also, most of them (64.44%) used EC within 72 hours after sexual intercourse to prevent unwanted pregnancies. The rest of them used EC within 24 hours after sex. Most of the respondents obtained EC from the pharmacy, over the counter (59.00%) followed by friends (25.52%). Only a few of them (15.48%) obtained EC as a form of prescription from the pharmacy (Table 4).

## Factors influencing the use of emergency contraception

Binary logistic regression analysis was conducted to identify the factors influencing the use of EC and presented in Table 5. The results showed that respondents from the Christian religion were 3.6 times more likely to use EC compared to those from the Islamic religion (95%CI (1.79–7.13), $p<0.001$). Also, respondents whose highest level of education was Senior high

**Table 4. Use of emergency contraceptives among women of reproductive age in the Kwadaso Municipality.**

| Variables | Frequency | Percentage |
|---|---|---|
| **Have you ever used any form of emergency contraception** | | |
| Yes | 239 | 79.67 |
| No | 61 | 20.33 |
| **If yes, why did you use an emergency contraceptive? (n = 239)** | | |
| Because I experienced condom breakage or slippage | 19 | 7.95 |
| Because I experienced failed coitus interruptus | 59 | 24.69 |
| Because of miscalculation of the rhythm method | 18 | 7.53 |
| Because I had unexpected unprotected sex | 143 | 59.83 |
| **When did you use EC to effectively prevent pregnancy after sex (n = 239)** | | |
| Within 24 hours | 85 | 35.56 |
| Within 72 hours | 154 | 64.44 |
| **Where did you access the ECs (n = 239)** | | |
| Friends | 61 | 25.52 |
| At the pharmacy, over the counter | 141 | 59.00 |
| At the pharmacy, prescription only | 37 | 15.48 |

school and below were less likely to use EC compared to those whose highest level of education was above Senior high school (COR = 0.51 [95%CI (0.27–0.95)], $p$ = 0.036). Additionally, respondents who received monthly income not more than GH₵ 1000 (COR = 0.34 [95%CI (0.14–0.79)], $p$ = 0.013) were less likely to use EC compared to those who received more than GH₵ 1000. Furthermore, respondents who showed a positive attitude towards EC were 2.9 times more likely to use EC compared to those who showed a negative attitude towards EC (95%CI (1.57–5.16), $p$ = 0.001).

Presented in Table 6 are the results of the multiple regression analysis of factors influencing the use of EC adjusting for religion, education, monthly income, and attitude towards EC. Respondents religion (AOR = 4.56 [95%CI (1.60–12.96)], $p$ = 0.004), monthly income (AOR = 0.29 [95%CI (0.09–0.88)], $p$ = 0.030), and attitude towards EC (AOR = 8.52 [95%CI (3.16–22.96)], $p<0.001$) remained significant predictors of the use of EC.

## Discussion

The present study explored the factors influencing the use of emergency contraceptives (ECs) among women of reproductive age in the Kwadaso Municipality. Unlike other regular method of contraception, ECs are mostly used after unprotected sexual intercourse to prevent unintended pregnancy. Knowledge regarding the use of EC is necessary and beneficial for its effectiveness [1]. The findings of this study showed that the majority (96.2%) of the participants had heard about emergency contraception. This finding is similar to the findings of the studies conducted in Botswana [23] and Ethiopia [1]. However, this finding is comparatively higher than that of the studies conducted in Tamale, Ghana [20, 24]. Additionally, studies in Kwa-Zulu-Natal Province, South Africa (56.4%) [25], Arba Minch University, Southern Ethiopia (63%) [26], Adama Town, Ethiopia (38.7%) [27], and India (14.3%) [28] showed a comparatively low level of EC awareness among study participants. The differences in the level of awareness could be due to differences in the study settings and time variation in relation to the acceleration of reproductive health promotion activities [29].

Progesterone-only-pill (92%) was the most commonly mentioned method of EC in this study followed by combined oral contraceptive pill (58.7%) and intrauterine copper device (54.67%) among the participants who had heard of EC in this study. Inconsistent with this

**Table 5. Factors influencing emergency contraception usage among the women in the Kwadaso Municipality.**

| Variables | Ever used EC | | COR (95%Cl) | P-value |
|---|---|---|---|---|
| | Yes (%) | No (%) | | |
| **Age** | | | | |
| ≤ 19 years | 19 (7.95) | 0 (0.00) | 1 (Ref) | |
| 20 to 24 | 114 (47.70) | 24 (39.34) | 1.34 (0.63, 2.86) | 0.446 |
| ≥ 25 | 106 (44.35) | 37 (60.66) | 0.71 (0.32, 1.54) | 0.381 |
| **Religion** | | | | |
| Muslim | 25 (10.46) | 18 (29.51) | 1 (Ref) | |
| Christianity | 214 (89.54) | 43 (70.49) | 3.58 (1.79, 7.13) | <0.001*** |
| **Highest level of education** | | | | |
| Above Senior high school | 141 (59.00) | 45 (73.77) | 1 (Ref) | |
| Senior high school and below | 98 (41.00) | 16 (26.23) | 0.51 (0.27, 0.95) | 0.036** |
| **Marital Status** | | | | |
| Married | 77 (32.22) | 25 (40.98) | 1 (Ref) | |
| Single | 162 (67.78) | 36 (59.02) | 0.68 (0.38, 1.21) | 0.198 |
| **Did you work for pay in the last month?** | | | | |
| No | 101 (42.26) | 29 (47.54) | 1 (Ref) | |
| Yes | 138 (57.74) | 32 (52.46) | 1.23 (0.70, 2.17) | 0.458 |
| **Monthly income (n = 170)** | | | | |
| > 1000 cedis | 63 (45.65) | 12 (37.50) | 1 (Ref) | |
| ≤ 1000 cedis | 75 (54.34) | 20 (62.50) | 0.34 (0.14, 0.79) | 0.013* |
| **Knowledge about EC** | | | | |
| Poor knowledge | 97 (40.59) | 17 (27.87) | 1 (Ref) | |
| Good knowledge | 142 (59.41) | 44 (72.13) | 0.57 (0.31,1.05) | 0.070 |
| **Attitude towards EC** | | | | |
| Negative attitude | 100 (41.84) | 41 (67.21) | 1 (Ref) | |
| Positive attitude | 139 (58.16) | 20 (32.79) | 2.85 (1.57, 5.16) | 0.001** |

Note: COR = Crude Odds Ratio; Ref = reference

Dummy variable; Ever used EC: yes = 1, no = 0

*p-value≤0.05

**p-value≤0.01

***p-value<0.001

finding is that of the study in South Africa where the authors reported combined oral contraceptive (82.5%) as the most commonly mentioned method of EC [25]. Most of the participants in this study had their source of information about EC from a health professional, followed by friends and media. In contrast, the main source of information about EC in other studies was the media [28, 30] and friends [5]. However, this finding is consistent with that of the study in Tamale, Ghana [20], Nigeria [31] and Ethiopia [1] where a higher proportion of participants indicated a health professional as their source of knowledge about EC. Also, 68% of the participants in this study knew the correct time frame to use EC. This finding is lower than the finding of the study conducted in Tamale (85%) [24] and in Arba Minch, Ethiopia (87.2%) [32], but higher than studies conducted in Botswana (38.2%) [23], Southern Ethiopia (51.5%) [26], Uttar Pradesh (54%) [33] and Nagpur district (46%) in India [30]. However, the study in Nigeria (66.7%) [31] and in Harar Town, Ethiopia (65%) [1] showed similar proportions with the present study. More so, in this study, 93% of participants knew the side effects of EC, and 89.3% of them could identify some side effects. This finding is similar to that of a

**Table 6. Multiple regression analysis of factors influencing emergency contraception usage among the women in the Kwadaso Municipality.**

| Predictor variables | Ever used EC | | AOR (95%Cl) | P-value |
|---|---|---|---|---|
| | Yes (%) | No (%) | | |
| **Religion** | | | | |
| Muslim | 25 (10.46) | 18 (29.51) | 1 (Ref) | |
| Christianity | 214 (89.54) | 43 (70.49) | 4.56 (1.60, 12.96) | 0.004** |
| **Highest level of education** | | | | |
| Above Senior high school | 141 (59.00) | 45 (73.77) | 1 (Ref) | |
| Senior high school and below | 98 (41.00) | 16 (26.23) | 0.97 (0.81, 1.17) | 0.776 |
| **Monthly income (n = 170)** | | | | |
| > 1000 cedis | 63 (45.65) | 12 (37.50) | 1 (Ref) | |
| ≤ 1000 cedis | 75 (54.34) | 20 (62.50) | 0.29 (0.09, 0.88) | 0.030* |
| **Attitude towards EC** | | | | |
| Negative attitude | 100 (41.84) | 41 (67.21) | 1 (Ref) | |
| Positive attitude | 139 (58.16) | 20 (32.79) | 8.52 (3.16, 22.96) | <0.001*** |

Note: AOR = Adjusted Odds Ratio; Ref = reference value

Dummy variable; Ever used EC: yes = 1, no = 0

*p-value≤0.05

**p-value≤0.01

***p-value<0.001

study conducted in Nigeria which reported that 72.5% of the participants had correct knowledge on the side effects of EC [31] but higher than that of the study in India (8.6%) [30]. In addition, 51.4% of the participants, who identified some side effects, noted menstrual irregularity. This finding is consistent with that of the study in Nagpur District, Ethiopia where the authors reported menstrual cycle irregularity as the side effect noted by most of the respondents [30]. Conversely, the study by Sibanda and Titus (2017) reported that most participants (90.3%) noted nausea and vomiting as a side effect of EC [25]. Similarly, 61.3% of the participants in this study noted that EC was not recommended as a regular contraceptive method. This is similar to the findings of Habitu et al. (2018) who reported that 58.6% of the participants felt that EC should not be used regularly [26]. However, 28.4% of the participants in the study conducted by Fekadu (2017) noted that ECs can be used as a regular contraceptive [32], which is higher than that of this study (16%). Research has indicated that poor publicity of contraceptives and limited availability are possible reasons for low awareness [34]. Thus, the high level of awareness of EC in the present study could be due to high publicity on the benefit of utilizing EC in the Municipality by the health workers through safe sex health promotion or awareness campaigns, since most of the participants indicated health professionals as the major source of information on ECs. Likewise, their awareness of the correct time frame to use ECs could be attributed to their source of information, that is, from health professionals. Also, the increased awareness of progesterone-only-pills could be likely due to its wide availability and patronage compared to the other ECs.

Generally, this study showed that the majority (62%) of those who had heard of EC had a good knowledge about EC. This finding is consistent with the finding of the study at Debre-Markos University, Ethiopia [29], however lower than studies conducted in India [30], and another part of Ethiopia [1, 32]. The good level of knowledge in this study could be attributed to the participants' source of information about EC as most of them were informed by a health professional. This implies that the health professionals in the Kwadaso Municipality are involved in effective reproductive health education and promotion.

Over half of all respondents (53%) had a positive attitude towards EC use. This finding is similar to that of a study done in Debre-Makos University in Ethiopia (53.8%) [29] but much lower than other studies also done in Ethiopia which reported 87.1% and 61.3 [27, 32]. The variations in the level of attitudes could be due the difference in the sociodemographic background of respondents, with younger respondents and students likely to a higher prevalence of positive attitudes.

Also, 48.67% of the participants in this study believed that the provision of EC to women would encourage promiscuity and hence increase the prevalence of HIV / AIDS and other STIs. Fifty-two percent (52%) of them agreed that the provision of EC would discourage compliance with other contraceptive methods, and 83.67% believed that repeated use of EC poses a health risk. The finding is similar to studies in other parts of Ghana [20], Ethiopia [2, 27], and South Africa [25]. This finding suggests the need to intensify community-based health education programs targeted at clarifying negative attitudes towards the use of emergency contraception [20].

Considering the use of EC, 79.7% of the participants in this study had ever used any form of EC. This finding is higher than that of the studies in Tamale, Ghana [20, 24]. Similarly, this finding is higher than studies in other countries like Botswana (22.0%) [23] and Ethiopia (58.8%) [32]. The higher rate of EC use among women in this study could be due to their higher knowledge and/or positive attitude which may be significant determinants of the use of ECs. Also, to further increase EC usage rate, health authorities should make ECs available and easily accessible to women of reproductive age.

More so, majority (59.8%) of those who had used EC cited unprotected sexual intercourse as the reason for using EC. Consistent with this finding in the study by Jima et al. (2017) in Adama, Ethiopia [27]. However, this finding disagrees with the study in Tamale where condom breakage or slippage was the main reason for using EC [20]. Also, 64.4% of the participants in this study used EC correctly, that is within 72 hours of sexual intercourse. Inconsistent with this finding, the study by Babatunde et al. (2016) found that most (85.7%) of the respondents in the public secondary school in Ilorin, Nigeria, used EC incorrectly after sexual intercourse [31]. Furthermore, 59% of the participants in this study obtained EC at the pharmacy, over the counter, followed by friends (25.5%). Only a few (15.5%) of them obtained EC at the pharmacy, with prescription only. Besides, none of them obtained EC at the local clinic with a prescription. This finding agreed with that of the study in Tamale, Ghana, where the pharmacy was the major source of EC for the respondents [20]. Family planning regulation suggests that ulipristal acetate EC pills should be obtained with prescription-only whether at the pharmacy or at local clinic [35]. In addition, to prevent any further health complications with the use of EC, pharmacists have been included as point sources of access and allowed to prescribe EC to women in the community. Thus, pharmacists can prescribe and provide EC directly to women [35]. This could be the reason for the procurement of EC at the pharmacy among the participants in this study.

This study further investigated the factors influencing EC use among reproductive-age women in the Kwadaso Municipality. The study found religion, monthly income and attitude towards EC as the main factors influencing EC use. Considering religion, the participants from the Christian religion were 4.6 times more likely to use EC compared to those from the Islamic religion. The low usage of EC among those from the Islamic religion could be due to the belief and greater opposition to contraceptive usage among Muslims [36]. Similarly, the study in Tamale, Ghana found a significant association between the use of EC and religion [20]. Regarding monthly income, this study found that participants with income less than or equal to GH₵1000 were less likely to use EC compared with those who earned above GH ₵1000. Furthermore, the study showed that participants who showed positive attitudes

towards EC were 8.5 times more likely to use EC than those who showed negative attitudes. This finding is consistent with that of the studies in Arba Minch University, Ethiopia [26, 32] where the authors found a significant association between attitude and the use of EC.

## Strengths and limitations

The study setting–Kwadaso Municipal is an urban area, therefore it is assumed that the study findings may not reflect the true nature of participants from rural communities of Ghana. Also, due to the sensitive nature of some of the questions, there may have been some bias in participants' responses in order to appear socially acceptable. However, it is hoped that the assurance of confidentiality and anonymity encouraged all participants to answer as truthfully as possible. Additionally, the study type–quantitative study, limited participants' responses, hence, future research should focus on using qualitative research approach to explore the predictors of EC use as well as widen the scope of research to gain more insight on this important subject matter in reproductive health. Besides, this study is the first of its kind in the Kwadaso Municipality, hence conducting the study among reproductive-age women at community level is novel and provides vital evidence that may be used as a basis for future decision making.

## Conclusions

Emergency contraceptive (EC) usage was high among women of reproductive-age in the Kwadaso Municipality. Most women had a good knowledge about EC and showed a positive attitude towards using EC in the future if the need be. Women's attitude towards EC, religion and monthly income were the major factors influencing the usage of EC in the municipality. Despite the positive attitude shown toward EC, most of the women believed EC could encourage sexual promiscuity and hence increase STIs. Some also indicated that increase use of EC discourages the use of regular modern contraceptives. These negative beliefs about EC require focused strategies and enhanced social communication to provide education on the proper use of EC in reducing unwanted pregnancies, as well as the role of other approaches in preventing STIs and improving overall reproductive health. This strategy will reduce the misconception regarding the use of emergency contraceptives, and create a conducive environment for other possible interventions.

## Supporting information

**S1 Fig.**
(XLSX)

**S1 File. Data collection tool.**
(DOCX)

**S2 File. Participant consent form.**
(DOCX)

**S1 Dataset.**
(XLSX)

## Author Contributions

**Conceptualization:** Daniel Sarpong Yeboah, Maxwell Afranie Appiah.

**Data curation:** Maxwell Afranie Appiah.

**Formal analysis:** Maxwell Afranie Appiah.

**Funding acquisition:** Daniel Sarpong Yeboah.

**Investigation:** Daniel Sarpong Yeboah, Maxwell Afranie Appiah, Grace Billi Kampitib.

**Methodology:** Daniel Sarpong Yeboah, Maxwell Afranie Appiah.

**Project administration:** Daniel Sarpong Yeboah.

**Resources:** Daniel Sarpong Yeboah, Maxwell Afranie Appiah, Grace Billi Kampitib.

**Software:** Maxwell Afranie Appiah.

**Supervision:** Grace Billi Kampitib.

**Validation:** Maxwell Afranie Appiah.

**Visualization:** Maxwell Afranie Appiah.

**Writing – original draft:** Maxwell Afranie Appiah.

**Writing – review & editing:** Daniel Sarpong Yeboah, Maxwell Afranie Appiah.

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
