## [Decision Letter · Decision Letter 0]

5 Aug 2021

PONE-D-21-19143

Factors influencing the use of emergency contraception among reproductive age women in the Kwadaso Municipality, Ghana

PLOS ONE

Dear Dr. Appiah,

Thank you for submitting your manuscript to PLOS ONE. After careful consideration, we feel that it has merit but does not fully meet PLOS ONE’s publication criteria as it currently stands. Therefore, we invite you to submit a revised version of the manuscript that addresses the points raised during the review process.

We look forward to receiving your revised manuscript.

Kind regards,

Imran Masood, B. Pharm, MBA, CQRM, PhD

Academic Editor

PLOS ONE

2. You indicated that you had ethical approval for your study. In your Methods section, please ensure you have also stated whether you obtained consent from parents or guardians of the minors (< 18)included in the study or whether the research ethics committee or IRB specifically waived the need for their consent.

3. Please include additional information regarding the survey or questionnaire used in the study and ensure that you have provided sufficient details that others could replicate the analyses. For instance, if you developed a questionnaire as part of this study and it is not under a copyright more restrictive than CC-BY, please include a copy, in both the original language and English, as Supporting Information. If the original language is written in non-Latin characters, for example Amharic, Chinese, or Korean, please use a file format that ensures these characters are visible.

4. Please state whether you validated the questionnaire prior to testing on study participants. Please provide details regarding the validation group within the methods section.

5. Please note that in order to use the direct billing option the corresponding author must be affiliated with the chosen institute. Please either amend your manuscript to change the affiliation or corresponding author, or email us at plosone@plos.org with a request to remove this option.

Additional Editor Comments (if provided):

Dear author/s

Both reviewers are mainly concerned about the methodology section of your manuscript especially regarding development and validation of data collection tools. They also seem unsatisfied about details in the methodology section which indicates linguistic weakness in the writeup. Please update/ correct your manuscript and give satisfactory answers to the questions raised by the reviewers.

I will also suggest getting your manuscript reviewed for English language from a professional person.

Reviewers' comments:

Reviewer's Responses to Questions

**Comments to the Author**

1. Is the manuscript technically sound, and do the data support the conclusions?

Reviewer #1: Partly

Reviewer #2: Yes

2. Has the statistical analysis been performed appropriately and rigorously? 

Reviewer #1: No

Reviewer #2: Yes

3. Have the authors made all data underlying the findings in their manuscript fully available?

Reviewer #1: No

Reviewer #2: Yes

4. Is the manuscript presented in an intelligible fashion and written in standard English?

Reviewer #1: Yes

Reviewer #2: No

5. Review Comments to the Author

Reviewer #1: The study entitled “Factors Influencing the Use of Emergency Contraception among Reproductive Age Women in the Kwadaso Municipality, Ghana” is very interesting and covers an important topic. I have the following concerns and questions regarding this study,

Abstract

• Kindly provide the key points of your methodology as the current text is hard to follow.

• In the result section, the authors should provide only their significant results.

Introduction

• Authors should write precisely why they are conducting this study?. A lot of published studies are available on this topic, kindly mention clearly what the current study will add.

Methods

• The authors did not provide references to existing literature from which they adopted the survey tool.

• Authors should also provide detail on how they validated their survey tool.

• Authors should discuss the questionnaire under a separate heading and should also provide it as a supplementary file.

• Authors should also provide details about their scoring criteria and on what basis they categorized knowledge and attitude of study participants.

Results

• The authors did not provide detail on which part of the variable (YES or No) they applied logistic regression analysis.

• The authors should also provide detail about the references used in logistic regression analysis. If “1” indicate the reference value then how there could be two reference values from one variable that is presented in table 5?.

Discussion

• The author should discuss the methodological limitations and strengths of the study more precisely.

Conclusion

• A more logical conclusion should be provided based on the results with significant importance and contribution of the study to the scientific literature. The results should not be repeated in conclusion.

Reviewer #2: 1. There are a number of typos and grammatical errors throughout the manuscript, e.g.,

(p2, line 63) “The IUDs on the hand can be effective when inserted…”

(P 3, line 122) “were select from...”

(P4, line 136) “through face-to-face interviews methods…”

(P11, line 291-92) “Unlike other regular contraceptives methods,…”

2. Authors are required to provide references from where they have adopted the questions for their questionnaire.

3. Explain the process of questionnaire translation into the local language, and, how its reliability and validity was examined?

4. Besides nausea and vomiting there are many other clinically significant untoward effects associated with ECPs, e.g. Menstrual irregularity, Breast tenderness and Ectopic pregnancy. Authors are required to discuss these as well in the introduction section of their manuscript, and discuss how these could influence ECPs use.

5. As ECPs cannot prevent the chances of sexually transmitted diseases, how this aspect could affect the person's attitude towards ECP use? secondly, the cost of ECPs must be explored for its use.

6. PLOS authors have the option to publish the peer review history of their article (what does this mean?). If published, this will include your full peer review and any attached files.

Reviewer #1: No

Reviewer #2: **Yes: **Dr. Allah Bukhsh

---

## [Author Response · Author response to Decision Letter 0]

23 Aug 2021

Dear Editors,

We, authors of the manuscript “Factors Influencing Emergency Contraception among Women of Reproductive Age in the Kwadaso Municipality, Ghana”, thank the reviewers for their generous comments and have edited the manuscript to address their concerns. 

Also, the style of the manuscript has been changed to meet PLOS ONE’s requirements, including those for file naming. All supporting documents noted in the manuscript have been duly provided.

We believe that the manuscript is now suitable for publication in your reputable journal.

Thank you.

Yours sincerely,

Maxwell Afranie Appiah

(Corresponding author)

RESPONSE TO ACADEMIC EDITOR AND REVIEWERS

RESPONSE TO ACADEMIC EDITORS

Comment 1: Please ensure that your manuscript meets PLOS ONE’s style requirements, including those for file naming.

Response: Manuscript has been updated to meet the PLOS ONE’s style requirements.

Comment 2: You indicated that you had ethical approval for your study. In your methods section, please ensure you have also stated whether you obtained consent from parents or guardians of the minors (<18) included in the study or whether the research ethics committee or IRB specifically waived the need for their consent.

Response: Statement regarding consent obtained from parents or guardians of the minors (<18) has been clarified on page 5 (line 191-192) of the revised manuscript. 

Comment 3: Please include additional information regarding the survey or questionnaire used in the study and ensure that you provided sufficient details that others could replicate the analyses. 

Response: Additional information regarding the questionnaire has been provided as a supporting information and indicated on page 18 on the revised manuscript.

Comment 4: Please state whether you validated the questionnaire prior to testing on study participants. Please provide details regarding the validated group within the methods section

Response: Details regarding questionnaire validation and validated group has been stated within the methods section on page 4 (147 -152) of the revised manuscript.

Comment 5: Please note that in order to use the direct billing option the corresponding author must be affiliated with the chosen institute. Please either amend your manuscript to change the affiliation or corresponding author, or email us at plosone@plos.org with a request to remove this option.

Response: Authors applied for the PLOS Global Participation Initiative (GPI) funding at time of manuscript submission.

RESPONSE TO REVIEWER 1

Abstract

Comment: Kindly provide the key points of your methodology as the current text is hard to follow

Response: The problem with methodology in the abstract has been clarified on page 1 of the revised manuscript.

Introduction

Comment: Authors should write precisely why they are conducting this study? A lot of published studies are available on this topic, kindly mention clearly what the current study will add.

Response: The purpose for conducting the study and its significants have been reported on page 3 (line 101 – 108) of the revised manuscript

Methods

Comment: The authors did not provide references to existing literature from which they adopted the survey tool.

Response: Reference to existing literature from which questionnaire was adopted as been reported on page 4 (line 146) of the revised manuscript.

Comment: Authors should discuss the questionnaire under a separate heading and should also provide it as a supplementary file.

Response: Questionnaire has been discussed under a separate heading. The questionnaire has been included as a supporting document in the manuscript.

Comment: Authors should also provide details about their scoring criteria and on what basis they categorized knowledge and attitude of the study participants.

Response: Details about the scoring criteria has been reported on page 5 (line 165 -178) of the revised manuscript.

Results

Comment: The authors did not provide detail on which part of the variable (YES or No) they applied the logistic regression analysis.

Response: Detail on the variable logistic regression analysis was applied has been reported on page 5 (line 178-181) of the revised manuscript.

Comment: The authors should also provide detail about references used in the logistic regression analysis. If “1” indicate the reference value then how there could be two reference values from one variable that is presented in table 5?

Response: Detail about reference used in logistic regression analysis has been reported. The reference value was represented with “1”, and the two reference values from one variable has been corrected.

Discussion

Comment: The author should discuss the methodological limitations and strengths of the study more precisely.

Response: Methodological limitations and strength have been precisely discussed.

Conclusion

Comment: A more logical conclusion should be provided based on the results with significant importance and contribution of the study tp the scientific literature. The results should not be repeated in conclusion

Response: Problem with conclusion has been clarified page 15 and 16 of the revised manuscript

RESPONSE TO REVIEWER 2

Comment 1: The are a number of typos and grammatical errors throughout the manuscript.

Response: All grammatical errors have been corrected.

Comment 2: Authors are to provide reference from where they have adopted the questions for the questionnaire.

Response: Reference to existing literature from which questionnaire was adopted as been reported on page 4 (line 146) of the revised manuscript.

Comment 3: Explain the process of questionnaire translation into the local language, and how it reliability and validity was examined?

Response: The process of questionnaire translation into local language has been reported on page 4 of the revised manuscript.

Comment 4: Besides nausea and vomiting there are many other clinically significant untoward effects associated with ECPs, eg. Menstrual irregularity, breast tenderness and ectopic pregnancy. Authors are required to discuss these as well in the introduction section of their manuscript, and discuss how these could influence ECPs use.

Response: Other clinically significant untoward effects associated with ECPs have been reported in the introduction section as required. 

Comment 5: As ECPs cannot prevent the chances of sexually transmitted diseases how could this aspect affect the person’s attitude towards ECP use? Secondly, the cost of ECPs must be explored for its use.

Response: The cost of ECPs has been reported on page 1 (line 68 – 69) of the revised manuscript. The study did not explore the relationship between EC use and sexually transmitted diseases. Sexually transmitted diseases were only captured as part of the knowledge assessment questions on ECs. We hope to explore the how STIs affects attitude towards EC use in future research.

---

## [Decision Letter · Decision Letter 1]

1 Dec 2021

PONE-D-21-19143R1Factors influencing the use of emergency contraception among reproductive age women in the Kwadaso Municipality, GhanaPLOS ONE

Dear Dr. Appiah,

Thank you for submitting your manuscript to PLOS ONE. After careful consideration, we feel that it has merit but does not fully meet PLOS ONE’s publication criteria as it currently stands. Therefore, we invite you to submit a revised version of the manuscript that addresses the points raised during the review process. Please submit your revised manuscript by Jan 15 2022 11:59PM. If you will need more time than this to complete your revisions, please reply to this message or contact the journal office at plosone@plos.org. Please include the following items when submitting your revised manuscript:A rebuttal letter that responds to each point raised by the academic editor and reviewer(s). You should upload this letter as a separate file labeled 'Response to Reviewers'.A marked-up copy of your manuscript that highlights changes made to the original version. You should upload this as a separate file labeled 'Revised Manuscript with Track Changes'.An unmarked version of your revised paper without tracked changes. You should upload this as a separate file labeled 'Manuscript'.If applicable, we recommend that you deposit your laboratory protocols in protocols.io to enhance the reproducibility of your results. Protocols.io assigns your protocol its own identifier (DOI) so that it can be cited independently in the future. For instructions see: https://journals.plos.org/plosone/s/submission-guidelines#loc-laboratory-protocols. Additionally, PLOS ONE offers an option for publishing peer-reviewed Lab Protocol articles, which describe protocols hosted on protocols.io. Read more information on sharing protocols at https://plos.org/protocols?utm_medium=editorial-email&utm_source=authorletters&utm_campaign=protocols.

We look forward to receiving your revised manuscript.

Kind regards,

Imran Masood, B. Pharm, MBA, CQRM, PhD

Academic Editor

PLOS ONE

Journal Requirements:

Additional Editor Comments:

Please address comments of reviewer #3 especially

Comment 3: Sampling or selection of one married women out of more than one in a selected household is done by simple random sampling. Clarify if it is random or convenient sampling as no explanation is given in this context?

Comment 4: In ‘Data Analysis’ section, authors have used 50% score as cut off value to describe knowledge as ‘poor’ or ‘good’ and attitude as ‘negative’ or ‘positive’. Provide the basis or reference for this interpretation.

Reviewers' comments:

Reviewer's Responses to Questions

**Comments to the Author**

1. If the authors have adequately addressed your comments raised in a previous round of review and you feel that this manuscript is now acceptable for publication, you may indicate that here to bypass the “Comments to the Author” section, enter your conflict of interest statement in the “Confidential to Editor” section, and submit your "Accept" recommendation.

Reviewer #1: All comments have been addressed

Reviewer #3: (No Response)

2. Is the manuscript technically sound, and do the data support the conclusions?

Reviewer #1: Yes

Reviewer #3: Partly

3. Has the statistical analysis been performed appropriately and rigorously? 

Reviewer #1: Yes

Reviewer #3: I Don't Know

4. Have the authors made all data underlying the findings in their manuscript fully available?

Reviewer #1: Yes

Reviewer #3: Yes

5. Is the manuscript presented in an intelligible fashion and written in standard English?

Reviewer #1: Yes

Reviewer #3: Yes

6. Review Comments to the Author

Reviewer #1: (No Response)

Reviewer #3: Comment 1: Authors need to limit the use of similar sentences through the manuscript.

Comment 2: Under the heading of introduction, it is mentioned that several studies have been conducted in Ghana regarding the factors influencing the use of EC in female college or university students. Authors need to add reference here.

Comment 3: Sampling or selection of one married women out of more than one in a selected household is done by simple random sampling. Clarify if it is random or convenient sampling as no explanation is given in this context?

Comment 4: In ‘Data Analysis’ section, authors have used 50% score as cut off value to describe knowledge as ‘poor’ or ‘good’ and attitude as ‘negative’ or ‘positive’. Provide the basis or reference for this interpretation.

Comment 5: It would be better if authors add a tabulated format under the heading of introduction describing the commonly used EC brands and their prices in Ghana instead of summing up in one sentence.

7. PLOS authors have the option to publish the peer review history of their article (what does this mean?). If published, this will include your full peer review and any attached files.

Reviewer #1: No

Reviewer #3: No

---

## [Author Response · Author response to Decision Letter 1]

22 Dec 2021

RESPONSE TO REVIEWER #3’s REVIEWS AND COMMENTS

Comment 1: Authors need to limit the use of similar sentences through the manuscript.

Response: The manuscript has been read thoroughly to remove similar sentences.

Comment 2: Under the heading of introduction, it is mentioned that several studies have been conducted in Ghana regarding the factors influencing the use of EC in female college or university students. Authors need to add reference here.

Response: References to the above statement have been added and reported on page 3 (line 100) of the revised manuscript

Comment 3: Sampling selection of one married woman out of more than one in a selected household is done by simple random sampling. Clarify if it is random or convenient sampling as no explanation is given in this context?

Response: Explanation to the selection of one woman from many in a household has been well clarified and been reported on page 4 (line 139-143) of the revised manuscript.

Comment 4: In ‘Data Analysis’ section, authors have used 50% score as cut off value to describe knowledge as “poor” or “good” and attitude as ‘negative’ or ‘positive’. Provide the basis or reference for this interpretation.

Response: Authors have revised the cut off point for scoring knowledge into ‘poor’ or ‘good’ and attitude into ‘negative’ or ‘positive’. This has been reported on page 5 (line 174 to 195). 

Authors have used the mean score cut off point to dichotomize knowledge and attitude level, and have given a thorough interpretation as well as provided reference to this effect. After this revision, all sections of the manuscript which were affected have been updated including the results, discussion, and abstract.

Comment 5: It would be better if authors add a tabulated format under the heading of introduction describing the commonly used EC brands and their prices in Ghana instead of summing up in one sentence.

Response: Authors acknowledge the reviewers concern regarding adding a tabulated format of EC brand and prices to the heading of the introduction. However, authors feel it will be more convenient to maintain the price ranges indicated in the manuscript for reasons that authors may not be able to provide reference for each price of EC brand. Also, most pharmacies in Ghana do not have price specific for EC brands due to the dollar rate instability.

Thank you.

---

## [Editor Report · Decision Letter 2]

15 Feb 2022

Factors influencing the use of emergency contraceptives among reproductive age women in the Kwadaso Municipality, Ghana

PONE-D-21-19143R2

Dear Dr. Appiah,

We’re pleased to inform you that your manuscript has been judged scientifically suitable for publication and will be formally accepted for publication once it meets all outstanding technical requirements.

Kind regards,

Imran Masood, B. Pharm, MBA, CQRM, PhD

Academic Editor

PLOS ONE

---

## [Editor Report · Acceptance letter]

18 Feb 2022

PONE-D-21-19143R2 

Factors Influencing the Use of Emergency Contraceptives among Reproductive Age Women in the Kwadaso Municipality, Ghana 

Dear Dr. Appiah:

I'm pleased to inform you that your manuscript has been deemed suitable for publication in PLOS ONE. Congratulations! Your manuscript is now with our production department. 

Kind regards, 

on behalf of

Dr. Imran Masood 

Academic Editor

PLOS ONE